# Study on Microstructure and Tribological Mechanism of Mo Incorporated (AlCrTiZr)N High-Entropy Ceramics Coatings Prepared by Magnetron Sputtering

**DOI:** 10.3390/nano14100814

**Published:** 2024-05-07

**Authors:** Jia Zheng, Yiman Zhao, Jingchuan Li, Sam Zhang, Jian Zhang, Deen Sun

**Affiliations:** 1Center for Advanced Thin Films and Devices, School of Materials and Energy, Southwest University, Chongqing 400715, China; 2School of Intelligent Manufacturing, Luoyang Institute of Science and Technology, Luoyang 471023, China; 3School of Aeronautics, Harbin Institute of Technology, Harbin 150001, China; 4Chongqing Chuanyi Control Valve Co., Ltd., Chongqing 400707, China

**Keywords:** high-entropy ceramics coatings, mechanical properties, tribological behavior

## Abstract

(AlCrTiZrMo_x_)N coatings with varying Mo content were successfully prepared using a multi-target co-deposition magnetron sputtering system. The results reveal that the Mo content significantly affects the microstructure, hardness, fracture toughness, and tribological behavior of the coatings. As the Mo content in the coatings increases gradually, the preferred orientation changes from (200) to (111). The coatings consistently exhibit a distinct columnar structure. Additionally, the hardness of the coatings increases from 24.39 to 30.24 GPa, along with an increase in fracture toughness. The friction coefficient is reduced from 0.72 to 0.26, and the wear rate is reduced by 10 times. During the friction process, the inter-column regions of the coatings are initially damaged, causing the wear track to exhibit a wavy pattern. Greater frictional heat is generated at the crest of the wave, resulting in the formation of a MoO_2_ lubricating layer. The friction reaction helps to reduce the shear force during friction, demonstrating the lower friction coefficient of the (AlCrTiZrMo_x_)N coatings. Both the hardness and fracture toughness work together to reduce the wear rate, and the (AlCrTiZrMo_x_)N coatings show excellent wear resistance. Most notably, although the columnar structure plays a negative role in the hardness, it contributes greatly to the wear resistance.

## 1. Introduction

As is well known, friction between components is the main cause of the failure of mechanical equipment under extreme conditions, such as high speed and high temperature [1,2]. Based on the data on global energy consumption and economic expenditure, energy losses related to friction-and-wear account for one-third of global energy consumption [3,4]. Traditionally, the way to reduce friction is usually to use oil or grease lubrication systems [5]. However, some devices with high cleanliness requirements do not allow the presence of foreign additives such as lubricating oil. Therefore, it is necessary to investigate new materials and lubricating media to reduce friction and wear.

Traditional friction theory holds that the strength and fracture toughness of materials determine the wear rate [6,7]. During the friction process, materials with high strength minimize wear by reducing the contact area, while high-toughness materials reduce the risk of brittle fracture owing to uniform plastic deformation [8,9]. Although hardness and toughness always rise and fall in a lever-like manner, they are equally important for the wear resistance of materials [6,10]. Banerjee et al. found that the coating with medium hardness showed the best wear resistance, while the coating with the highest hardness showed brittle characteristics [11]. Wang et al. found that the CrBCN-L coating with the lowest hardness (17.9 GPa) but the best toughness (K_IC_ = 2.81 MPa m^1/2^) showed the best wear resistance (wear rate = 2.4 × 10^−7^ mm^3^/(N·m)) [12]. Therefore, overcoming the tradeoff between the fracture toughness and hardness of materials has become an effective way of enhancing the wear resistance of materials. Furthermore, previous studies have shown that the oxide of Mo has a layered structure and is easy to shear, which helps to promote the wear resistance of the material [13,14].

In recent years, high-entropy alloy coatings (HEA) have attracted significant attention due to their unique design concept and excellent comprehensive performance [15,16]. On that basis, the high-entropy ceramics coatings (HEC) are formed by filling nitrogen atoms in the interstitial position of high-entropy alloy coatings, which process a stronger lattice distortion effect which leads to better mechanical properties [17,18]. Excellent mechanical properties make high-entropy ceramics coatings a potential candidate for wear-resistant materials [19].

The objective of this study is to investigate how the addition of the lubricating element Mo harmonizes the hardness and fracture toughness and thereafter affects the tribological properties of the material. To do that, a series of (AlCrTiZrMo_x_)N coatings with different Mo contents were deposited using magnetron co-sputtering. The elemental composition, microstructure, mechanical properties, and tribological properties of the coatings were investigated, and the relationship between the wear resistance and the mechanical properties of the coatings was revealed.

## 2. Experimental

### 2.1. Deposition of Coatings

A multi-target co-deposition magnetron sputtering system (M230, Vikaitech, Beijing, China) was used to prepare (AlCrTiZrMo_x_)N coatings with different Mo contents by adjusting the Mo target power. The substrates were selected as Si (111) wafers and SS316L stainless steel (20 mm× 20 mm× 2 mm). (Details can be found in [20].) The target materials were TiZr (equal moles, ~99.9% purity), AlCr (equal moles, ~99.9% purity), and Mo (~99.9% purity). During the deposition process, the targets were connected with a DC power source, and the radio frequency (RF) bias was maintained at a power of 30 W, the Ar flow rate was 40 sccm, and the N_2_ flow rate was 12 sccm. The deposition process was performed with varying Mo target powers of 0, 30, 50, and 100 W to prepare four different coating systems. During this process, no additional temperature is applied and the maximum temperature during deposition is below 80 °C. Once the base pressure in the sputtering chamber reached 6.67 × 10^−4^ Pa, Ar gas was inlet and plasma was generated by applying bias voltage. The targets and substrate surfaces were pre-cleaned by plasma to remove surface impurities. Among them, the target material was pre-sputtered for 20 min, and the substrate was plasma pre-cleaned for 30 min. After that, coating deposition was conducted. During the deposition process, the working pressure was maintained at 4.47 × 10^−1^ Pa, the RF bias power was 30 W, the Ar/N_2_ flow rate was 40/12 sccm, and the sample tray was rotated at a constant speed of 15 rpm for coating uniformity purposes. For specific target power parameters, please refer to Table 1. The transition layer of 60 nm was first deposited on the substrate, followed by a functional layer of about 1.3 μm.

### 2.2. Microstructure Characterization

A grazing incidence X-ray diffractometer (GIXRD, SmartLab3KW, Rigaku Corporation, Tokyo, Japan) was used to determine the crystal structure of the (AlCrTiZrMo_x_)N coatings. The GIXRD instrument utilized Cu Kα radiation with a working voltage of 40 kV and a working current of 40 mA. The scanning step size was set at 5°/min with a scanning range of 20–90° and an incident angle of 1°. Based on the maximum diffraction peak data, the crystallite size (*d*) and lattice constant (*a*) were calculated by the following equation:(1)d =0.89λ/βcosθ
where *β* is the full width at half maximum (FWHM), *λ* is the X-ray wavelength (1.5406  Å), and *θ* is the diffraction angle [21].
(2)$$a=\sqrt{h^{2}{+k}^{2}{+l}^{2}}\times \;(\lambda /2\sin\theta )$$
where (*hkl*) are the planes associated with the individual XRD peaks [22]. The surface, cross-sectional morphology, as well as chemical composition of the coatings were characterized by the field emission scanning electron microscope (FESEM, ZEISS Sigma 300, Carl Zeiss AG, Oberkochen, Germany) and energy dispersion spectrometer (EDS, JEOL-6300 F, JEOL Ltd., Tokyo, Japan), respectively. The surface morphology and roughness of the coatings were tested by atomic force microscopy (AFM, Dimension Icon, Bruker Corporation, Billerica, MA, USA). In tapping mode, the test size was 3 μm × 3 μm, and the root mean square roughness (Rq) was analyzed by Imager 4.7.18 software. To further reveal the microstructure and composition of the post-friction coating, the morphology and chemical composition of the wear track were characterized using FESEM. X-ray photoelectron spectroscopy (XPS, Thermo K-Alpha^+^, Thermo Fisher Scientific Inc., Waltham, MA, USA) with Al Kα radiation was employed to investigate the chemical binding of the coatings and the wear track. The C 1s peak at 284.8 eV was used for calibration. Detailed analysis of the coatings was performed using Avantage v5.9921 software.

### 2.3. Mechanical and Tribological Tests

The mechanical properties of the coatings were tested at room temperature using the Nano indenter G200 (Keysight Technologies, Santa Rosa, CA, USA) fitted with a Berkovich indenter. The tests were performed in quasi-static mode, and the hardness (H) and modulus (E) values were obtained by fitting the Oliver–Pharr model. To minimize the substrate effect, a pressing depth of 110 nm was selected, which is less than 1/10 of the coating thickness. Each coating was tested six times to ensure data reliability. The fracture toughness of the prepared coatings was evaluated using the Vickers indentation method, and the normal load during the tests was 1.4 N. The damage is qualitatively compared based on the indentation crack. Each sample was tested at least three times to reduce errors. A ball-on-disc tribometer (TRB^3^, Anton Paar, Graz, Austria) was used to evaluate the tribological properties of the (AlCrTiZrMo_x_)N coatings. The friction counterpart, amplitude, total sliding distance, normal load, and constant frequency are Al_2_O_3_ (Φ = 6 mm), 2.5 mm, 20 m, 1 N, and 1 Hz, respectively. After the tests, a 3D surface profilometer (DektakXT, Bruker, Billerica, MA, USA) was used to scan the cross-sectional and three-dimensional profile (3D) of the wear track. The cross-sectional data were imported into Origin 2021 (9.8) software for integration to calculate the wear rate, which was determined according to Equation (3) [23]:(3)W=V/FS
where *W* is the wear rate (mm^3^/(N·m)), *V* is the wear volume (mm^3^), *F* is the normal load (N), and *S* is the total sliding distance (m).

## 3. Results and Discussion

### 3.1. Microstructures

Figure 1 shows the elemental composition of (AlCrTiZrMo_x_)N coatings deposited with different Mo target powers. As shown by the curve, the Mo atomic percentage in the coatings increases from 0 to 20.54 at.% as the Mo target powers varies from 0 to 100 W, while the N atomic percentage remains similar at 50–60 at.%. The high content of N in the coatings is due to the strong affinity between each target element and N [24]. Additionally, slight variations in the elemental content of Ti, Zr, Al, and Cr are observed in the coatings due to the difference in sputtering and reflection rates between components [25].

The GIXRD patterns of (AlCrTiZrMo_x_)N coatings prepared with different Mo target powers are shown in Figure 2. The diffraction peaks at 36.0°, 42.2°, 60.9°, and 73.1°correspond to (111), (200), (220), and (311) crystal planes, respectively, which matches with the face-centered cubic (FCC) phase structure (B1-NaCl type), consistent with previous studies [18,26]. The coating with 0 at.% Mo shows a (200) preferred orientation, and with the addition of Mo, the preferred orientation of the coatings changes to a (111) crystal plane. In other words, the Mo content has a dramatic effect on the preferred orientation of the coatings. Based on Pelleg’s study on crystallographic texture using total energy minimization, the lowest surface and strain energy of the FCC structure are (200) and (111) crystal planes, respectively [27]. The preferred orientation of a material is determined by the competition between strain energy and surface energy. Thermodynamically, the coatings tend to grow along the low-strain or low-surface-energy planes. The coatings with the Mo element exhibit a preference for growth on the (111) crystal plane, which helps mitigate the increase in strain energy on account of the bombardment of high-energy Mo atoms [28,29]. Conversely, the coating without the Mo element prefers to grow on the (200) crystal plane to minimize the contribution of surface energy. Based on the GIXRD information, the lattice constants (a, Å) and crystallite size (d, nm) of each sample are further calculated (cf. Table 2).

To investigate the bonding situations of the surface elements in the coatings, core-level regions of Ti 2p, Zr 3d, Al 2p, Cr 2p, Mo 3d, and N 1s were measured by the high-resolution XPS spectra and were recorded in Figure 3. The peak positions of Ti 2p_3/2_, Zr 3d_5/2_, Al 2p_3/2_, Cr 2p_3/2_, and Mo 3d_5/2_ are located at 453.00 eV, 177.90 eV, 71.00 eV, 573.90 eV, and 227.40 eV, respectively, which are close to the standard reference position [30,31,32,33]. Correspondingly, the metal-nitride peaks are located at 455.10 eV (Ti-N) [34], 179.60 eV (Zr-N) [35], 73.50 eV (Al-N) [36], 574.50 eV (Cr-N) [10], and 228.53 eV (Mo-N) [37]. The observed deviation of the binding energy of most metal elements from the metallic state (as shown by the blue dashed line) to the metal–nitride state (as shown by the pink dashed line) can prove the formation of the ceramics coating. Additionally, the increasing relative peak intensity of Mo 3d in Figure 3e indicates an increasing Mo content in the coatings. That result is further supported by the continuously increasing volume fraction of the Mo 3p peak in Figure 3f. Similar phenomena have been reported in previous studies [38].

The cross-sectional FESEM images of (AlCrTiZrMo_x_)N coatings in Figure 4 reveal that the thickness of the 4 coatings ranged from 1.1 to 1.4 µm. The cross-sections of all coatings show a similar columnar structure (Figure 4(a1,b1,c1,d1)). In other words, the growth pattern of the coatings is not significantly affected by the Mo target power. The columnar structure of the coatings is due to the atomic shadowing effects [39]. Similar results have been observed in previous studies on VAlTiCrMo high-entropy ceramics coatings and AlCrTiZrHf high-entropy ceramics coatings [40,41]. The FESEM images of the coating surface reveal that an increase in Mo target power results in more prominent particle aggregation on the coating surfaces. That phenomenon is attributed to a decrease in the diffusivity of surface atomic species, promoting island-like growth [42].

Figure 5 displays the surface profile images by AFM of (AlCrTiZrMo_x_)N coatings with varying Mo contents. It is evident that all the coating surfaces exhibit a typical cauliflower-like morphology. Compared to the coatings containing the Mo element, the coating with 0 at.% Mo exhibits a smooth and dense surface, and the roughness of this coating is only 4.4 nm. With the increase in the Mo content, the surface roughness of the ceramics coatings gradually increases (Figure 5b–d), measuring the values of 5.1 nm, 5.9 nm, and 8.0 nm, respectively. The increase in the surface roughness can be put down to the growth rate of the coatings. Lai et al. have reported that in transition metal ceramics coatings, the diffusion of deposited atoms is more challenging on the (111) crystal plane compared to the (200) crystal plane [18]. Therefore, when the (111) crystal plane becomes a preferred orientation, rapid growth in the vertical direction of the grains is promoted [43].

### 3.2. Mechanical Properties

Figure 6a illustrates the variation in hardness and Young’s modulus with the Mo content in the coating. The hardness and Young’s modulus of the coating without the Mo element are measured as 24.39 and 317.91 GPa, respectively. With the addition of Mo, the hardness of the coating increases. For the coating with 12.32 at.% Mo, the hardness reaches a maximum value of 30.24 GPa. However, further increasing of the Mo content to 20.54 at.% results in the hardness of the coating decreasing. Simultaneously, the value of Young’s modulus fluctuates in a small range. Based on the values of the hardness and Young’s modulus, H/E and H^3^/E^2^ of the coatings were calculated, representing the elastic deformation ability and anti-plastic deformation ability of the coating, respectively, as shown in Figure 6b. It can be seen that the H/E and H^3^/E^2^ of the coating with 12.32 at.% Mo are 0.092 and 0.257, respectively, which are significantly higher than other coatings. It is predicted that the coating with 12.32 at.% Mo has the strongest crack propagation resistance and the best friction and wear performance.

For coatings prepared by physical vapor deposition (PVD), the hardness primarily relies on the effects of the grain boundaries and solid solution [44,45]. The grain boundary strengthening and solution strengthening are the main factors to improve the hardness of the PVD coatings. Firstly, the crystallite size of the (AlCrTiZrMo_x_)N coatings, calculated from the strongest peak of the GIXRD pattern, is found to be smaller than 10 nm (cf. Table 2). With the increase in the Mo content from 0 to 6.26 at.%, the crystallite size rises from 6.5 to 7.1 nm. When the Mo content increases further, the crystallite size decreases gradually. The crystallite size of the coatings follows the reverse Hall–Petch effect [6], which depicts a larger crystallite size leading to more pronounced strengthening effects when the crystallite size is less than 10 nm. Secondly, the addition of Mo increases the lattice constant of the coatings (cf. Table 2), and the expansion of the cubic lattice results in solution strengthening. As for (AlCrTiZrMo_x_)N coatings, it is further speculated that solution strengthening plays a predominant role, ultimately contributing to the highest hardness in the coating of 12.32 at.% Mo. In conclusion, the enhanced hardness of the coatings can be attributed to the strengthening effects of the grain boundary and solid solution.

The surface morphology of the coatings at the indentation site was examined using FESEM, as illustrated in Figure 7. The sample surface appears smooth and flat, with radial cracks propagating outward along the diagonal of the indentation. Notably, the coating with 0 at.% Mo has the longest radial crack length, and localized cracks are observed within the indentation pits. In contrast, the Mo-containing coatings exhibit significantly shorter radial cracks, especially for the coating with 12.32 at.% Mo, where no significant radial crack is observed. This indicates that (AlCrTiZrMo_x_)N coatings possess higher toughness, which impedes crack propagation and enhances energy dissipation. A similar phenomenon has been observed in previous studies [46,47].

### 3.3. Friction and Wear Properties

#### 3.3.1. Friction Coefficient and Wear Rate

Figure 8a shows the correlation between the friction coefficient (COF) and the wear distance of the coatings. It is evident that all coatings demonstrate an initial break-in period, followed by a stabilization of the friction coefficient. The coating with 0 at.% Mo exhibits a high friction coefficient of 0.78 with significant fluctuations, whereas the Mo-containing coatings display a lower coefficient and higher stability. In particular, the coating with 12.32 at.% Mo achieves a low friction coefficient of 0.26. Additionally, noticeable distinctions in the friction behavior between (AlCrTiZrMo_x_)N coatings and (AlCrTiZr)N coatings can be observed during the initial break-in period. At the initial running-in period, the friction coefficient of (AlCrTiZrMo_x_)N coatings increases rapidly, which is considered to be related to surface roughness. In the sliding process, the larger surface roughness leads to the greater friction of the friction counterpart. When the friction distance is 2–3 m, the friction coefficient of the (AlCrTiZrMo_x_)N coatings is reduced, and the running-in period is over. This indicates that the formation of a shear-ready friction layer and the initiation of a stable friction regime.

Figure 8b depicts the friction coefficients and wear rates of the coatings, accompanied by a 2D cross-sectional wear track image. As shown, the friction coefficient and wear rate present the same trend, decreasing first and then increasing. When the Mo content is 0, 6.26, 12.32, and 20.54 at.%, the friction coefficients of the coatings are 0.72, 0.35, 0.26, and 0.38, respectively. Calculate the wear rate of the coatings to further analyze their tribological properties. The (AlCrTiZrMo_x_)N coatings are an order of wear rates less massive than the (AlCrTiZr)N coating. Notably, the coating with 12.32 at.% Mo achieves the lowest wear rate (3.15 × 10^−6^ mm^3^/(N·m)).

#### 3.3.2. Wear Track Morphologies

In order to further explore the wear condition of the coatings, the surface morphologies and 3D morphologies of the coatings are shown in Figure 9. The wear surface of the coating with 0 at.% Mo appears as thin grooves parallel to the sliding direction, accompanied by pits and attachments (cf. Figure 9(a,a1)), which exhibit typical adhesive wear and abrasive wear, indicating severe wear. The wear surface of the coating with 6.26 at.% Mo shows wide grooves and a small number of debris (cf. Figure 9(b,b1)). In contrast, the coating with 12.32 at.% Mo displays broad and smooth grooves on the wear surface (cf. Figure 9(c,c1)). Similarly, the wear surface of the coating with 20.54 at.% Mo shows grooves and a few cracks (cf. Figure 9(d,d1)). The latter three coatings demonstrate significant abrasive wear. These observations suggest the existence of distinct wear mechanisms between the (AlCrTiZr)N and (AlCrTiZrMo_x_)N coatings. Furthermore, the wear depths of the coatings with 0, 6.26, 12.32, and 20.54 at.% Mo are 982 nm, 823 nm, 740 nm, and 847 nm, respectively. The wear depth and width (marked in Figure 9) of the (AlCrTiZrMo_x_)N coatings are significantly lower than that of the (AlCrTiZr)N coating, indicating an obvious improvement in wear resistance of coatings with the incorporation of the Mo element. The coatings present a columnar structure (cf. Figure 4), and the inter-columnar areas act as a weak area, in which it is easy to slide between the columns during the friction process [48,49]. That inter-columnar sliding leads to the formation of a wavy wear pattern on the wear surface, which effectively reduces the wear. However, the coating with 0 at.% Mo, despite possessing a columnar microstructure, exhibits completely different wear morphologies. This discrepancy motivates further investigation of the specific contribution of Mo in the coatings.

## 4. Tribological Mechanism

### 4.1. Element Analysis of Wear Track

So as to further investigate the influence of Mo on the friction of the (AlCrTiZrMo_x_)N coatings, the element mapping and chemical bonding of the wear surfaces were analyzed. Firstly, the element distribution on the wear surfaces is illustrated in Figure 10. In the case of the coating with 0 at.% Mo, the signal of the substrate element (Fe) is clearly visible, while there are no distinct signals indicating the presence of O or other elements. On the contrary, no obvious substrate element signal is observed on the wear track of the coating with 12.32 at.% Mo. Meanwhile, the O element signal shows a streaky aggregation pattern, corresponding to the wave crest observed in the 3D wear morphologies (cf. Figure 9), which indicates that the frictional heat is generated between the friction counterpart and the coating, leading to oxidation. Consequently, the oxide layer adheres to the surface of the coating, reducing the shear force generated during the friction process, thereby lowering the friction coefficient and slowing down the wear.

Figure 11 shows the survey spectra of the pre/post-friction test surfaces, as well as the fine spectra of high-resolution core level peaks for Ti 2p, Zr 3d, Al 2p, Cr 2p, and Mo 3d. The high-resolution XPS spectra of the coating wear surfaces were further analyzed to examine the extent of oxidation. Prior to the coating deposition, although the base pressure was evacuated to 6.67 × 10^−4^ Pa, there was still some residual oxygen in the deposition chamber. Therefore, the presence of oxygen-containing chemical bonds can be observed in both pre-friction samples. In the Ti 2p spectra (cf. Figure 11(b,b1)), the 2p_3/2_ and 2p_1/2_ peaks are deconvoluted into six chemical binding states, which correspond to Ti-N bonds (bonding energy (BE) = 455.10 eV, 460.80 eV) in TiN, Ti-N-O bonds (BE = 456.80 eV, 462.50 eV) in TiN_x_O_y_, and Ti-O bonds (BE = 458.30 eV, 464.00 eV) in TiO_2_. The Zr 3d spectra are decomposed into four bonding states (cf. Figure 11(c,c1)), corresponding to Zr-N bonds in ZrN (BE = 179.60 eV, 181.95 eV) and Zr-O bonds in ZrO_2_ (BE = 181.40 eV, 183.55 eV). The deconvoluted Al 2p spectra shown in Figure 11(d,d1) exhibit two bonding states, attributed to the Al-N bond (73.50 eV) in AlN and Al-O bond (74.20 eV) in Al_2_O_3_. Similarly, the Cr 2p spectra (cf. Figure 11(e,e1)) show two main peaks at 574.50 eV and 583.82 eV, corresponding to the Cr-N bond in CrN and peaks at 576.07 eV and 585.63 eV, indicative of the Cr-O bond in Cr_2_O_3_. The Mo 3d spectra (cf. Figure 11(f1)) also identify the Mo-N bond in MoN (BE = 228.53, 231.73 eV) and Mo-O bond in MoO_2_ (BE = 229.80, 231.85 eV), respectively. Table 3 summarizes the atomic ratios of the pre/post-friction of the metal oxides for the coatings with 0 at.% Mo and 12.32 at.% Mo. During the process of friction, both coatings produce a friction oxide layer, which contains TiO_2_, Al_2_O_3_, and Cr_2_O_3_. Moreover, the presence of MoO_2_ on the wear surface of the coating with 12.32 at.% Mo, which acts as a friction layer, is prone to shearing, providing lubrication and effectively reducing the friction coefficient. Additionally, the generated oxide layer covers the wear surface, effectively isolating contact between the coating and the friction counterpart, providing protection. Therefore, in addition to the adhesive and abrasive wear analyzed above, the wear mechanism of the coating also includes oxidative wear.

Typically, the tribological performance of the coatings is significantly affected by the transfer film formed on the friction counterpart. Figure 12 illustrates the distinct transfer film morphologies on the contact surface of friction counterparts. In the case of the coating with 0 at.% Mo, a substantial amount of thick block-like transfer material is observed on the contact surface of the friction counterpart, which corresponds to a higher wear rate of the coating. Conversely, the coating with 12.32 at.% Mo exhibits a thin-layer transfer film on the contact surface. It can be inferred from the EDS spectra that the transfer film of the coating with 0 at.% Mo mainly consists of elements such as Ti and Cr, while the transfer film of the coating with 12.32 at.% Mo mainly consists of elements such as Ti, Cr, and Mo. Consequently, the friction process occurs between the coating and the transfer film dominated by the Mo element, resulting in a lower friction coefficient and wear rate. (In this qualitative description, the results only consider Ti, Cr, and Mo elements, as Al and O are constituents of the friction counterpart, and Zr is similar to the Pt element sprayed during the testing process.)

### 4.2. Wear Mechanism

Clearly, the friction test shows that the (AlCrTiZrMo_x_)N coatings exhibit enhanced tribological properties. Figure 13 summarizes the wear mechanism of (AlCrTiZrMo_x_)N high-entropy ceramic coatings. This is primarily reflected in two aspects: the lower friction coefficient and the wear rate. The reason for the low friction coefficient is the formation of MoO_2_ due to friction heat, and the layered oxide effectively reduces the friction coefficient. In the case of the (AlCrTiZr)N coating, metallic oxides are uniformly distributed along the wear track. However, for the (AlCrTiZrMo_x_)N coating, the distribution of layered oxide has obvious segregation, and the content of oxides is higher at the peak of the wear track, which effectively reduces the shear force. Moreover, the low wear rate is put down to the synergistic effect of hardness and fracture toughness. For the (AlCrTiZr)N coating, the lower hardness leads to poor resistance against damage, while low toughness causes the wear mode characterized by fracture. Cracks initiate in the inter-columnar regions and propagate, leading to extensive fracture and severe wear along the wear track. However, for the (AlCrTiZrMo_x_)N coatings, both the hardness and fracture toughness are enhanced with the incorporation of the Mo element. In particular, the improvement in toughness shifts the wear mode from fracture to plastic deformation. The cracks generated in the inter-columnar regions have limited propagation, and fracturing occurs in partial areas of the wear track. With the progression of friction, the segregation of oxides at the crest of the wear track further effectively decreases fracturing. Based on these aspects, (AlCrTiZrMo_x_)N coatings exhibit excellent tribological properties.

## 5. Conclusions

In this work, the (AlCrTiZrMo_x_)N high-entropy ceramics coatings with diverse Mo contents were successfully prepared by the DC magnetron sputtering technique. The correlation between microstructure, mechanical properties, and tribological properties of the deposited coatings was studied. The following conclusions can be drawn:GIXRD results show that the prepared coatings are all solid solution structures. With the addition of the Mo element, the crystallinity increases and the preferred orientation of the coating changes from (111) to (200). In addition, the coating with the (111) preferred orientation presents higher roughness.Hard yet tough high-entropy ceramics coatings (TiZrAlCrMo_x_)N can be obtained by magnetron sputtering, with a hardness of 30.24 GPa, and toughness (no obvious propagation cracks appear).The coatings with Mo show excellent tribological properties compared to those without Mo: the friction coefficient reduces to 1/3 (from 0.72 to 0.26) and the wear rate reduces to 1/10 (from 3.39 × 10^−5^ to 3.15 × 10^−6^ mm^3^/(N·m)).The low friction coefficient results from the formation of layered MoO_2_ which reduces the shear force, and the low wear rate results from both the high hardness and toughness.

## Figures and Tables

**Figure 1 nanomaterials-14-00814-f001:**
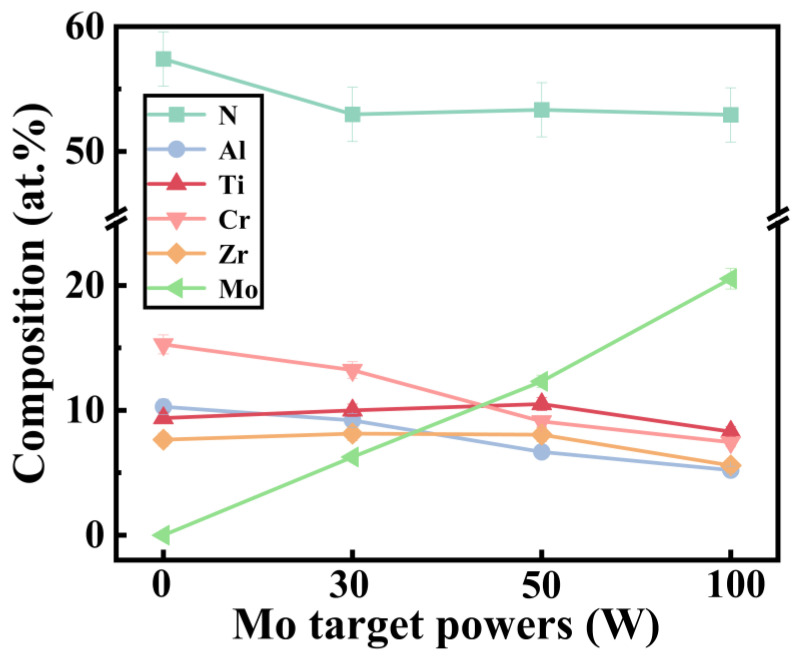
Elemental composition (by EDS) of the (AlCrTiZrMo_x_)N high-entropy ceramics coatings deposited with different Mo target powers.

**Figure 2 nanomaterials-14-00814-f002:**
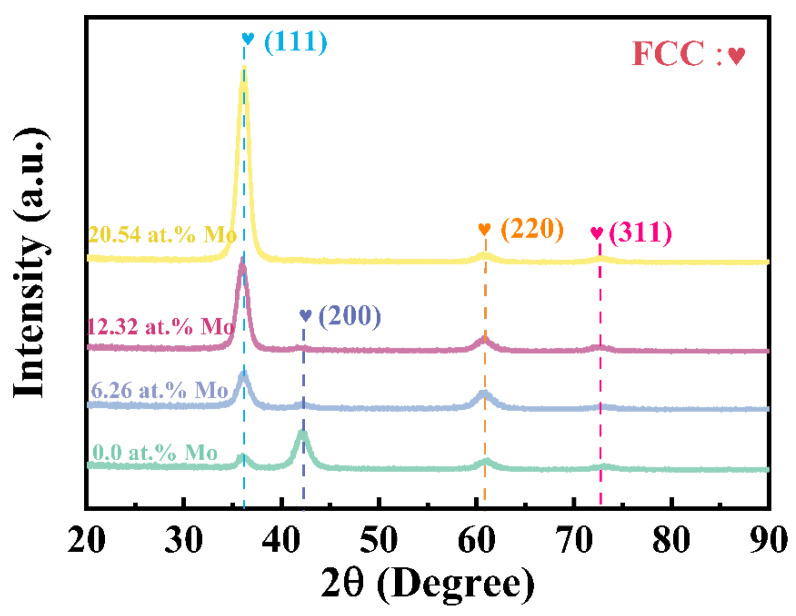
GIXRD patterns of (AlCrTiZrMo_x_)N coatings deposited at different target powers.

**Figure 3 nanomaterials-14-00814-f003:**
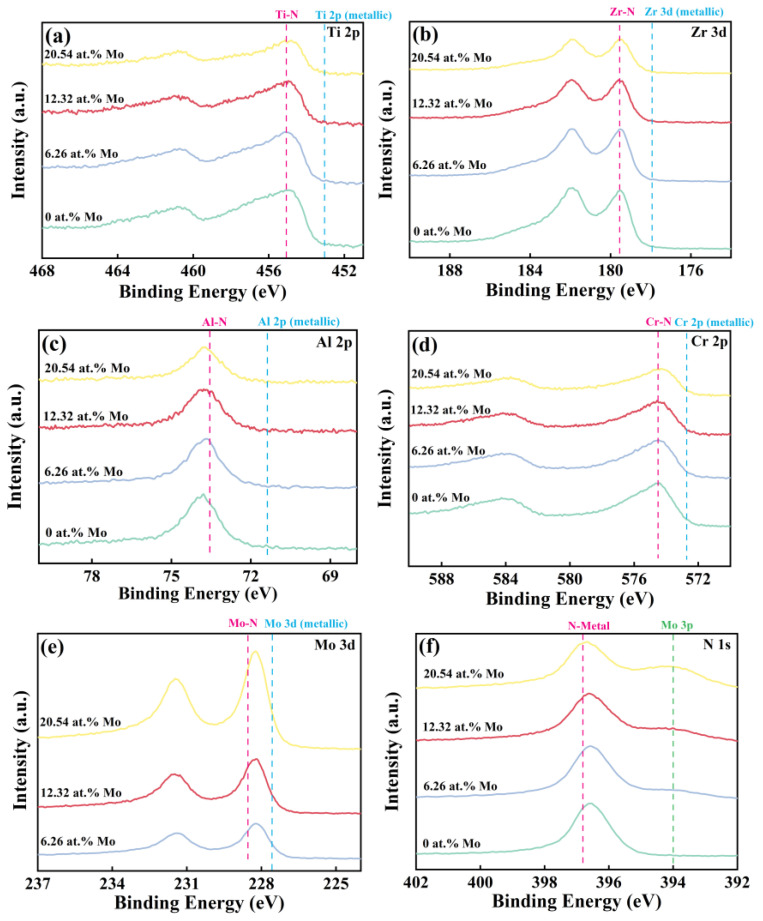
XPS spectra of (AlCrTiZrMo_x_)N high-entropy ceramics coatings, showing the core energy levels spectra of (**a**) Ti 2p, (**b**) Zr 3d, (**c**) Al 2p, (**d**) Cr 2p, (**e**) Mo 3d, and (**f**) N 1 s.

**Figure 4 nanomaterials-14-00814-f004:**
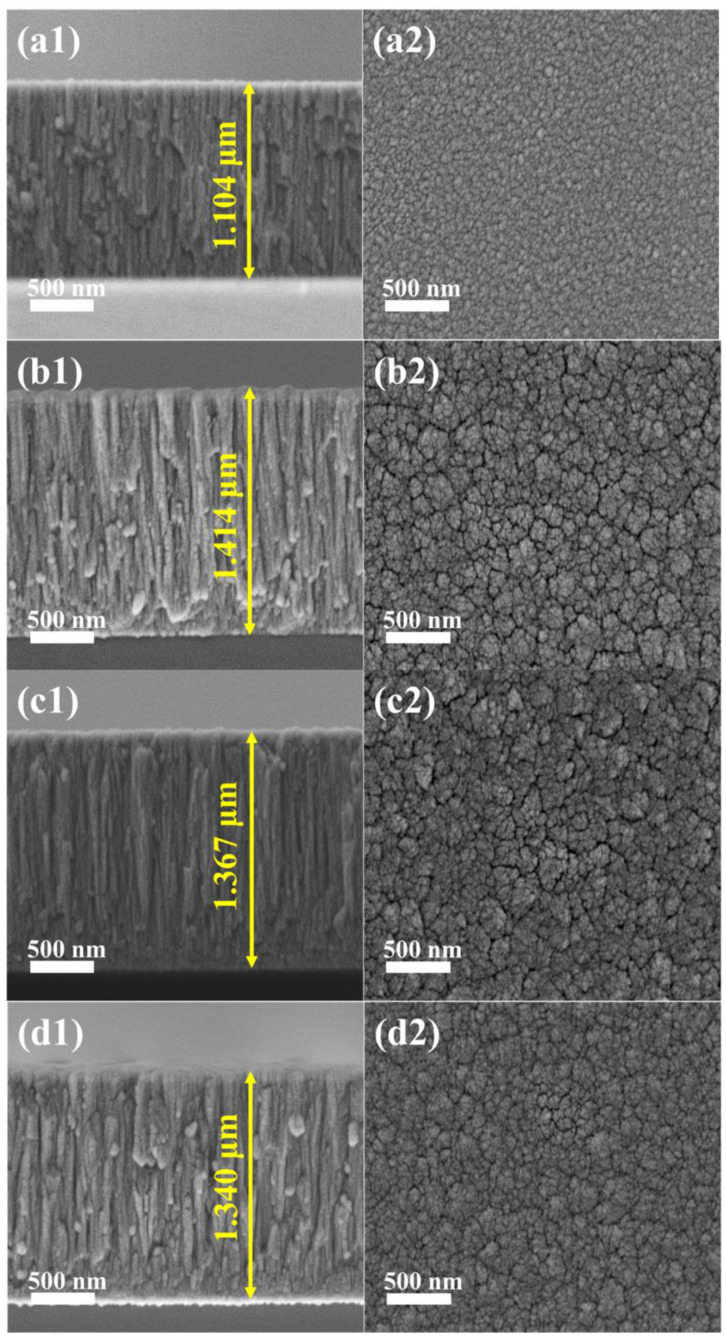
The FESEM cross-section and surface images of the (AlCrTiZrMo_x_)N coatings with different at.% Mo: (**a1**,**a2**) 0, (**b1**,**b2**) 6.26, (**c1**,**c2**) 12.32, and (**d1**,**d2**) 20.54.

**Figure 5 nanomaterials-14-00814-f005:**
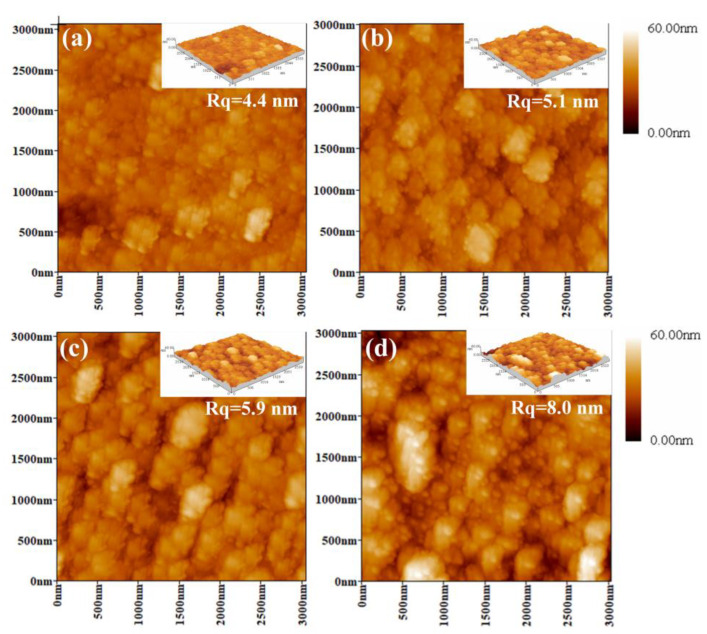
AFM micrographs (3 μm × 3 μm scans) of (AlCrTiZrMo_x_)N coatings with different at.% Mo: (**a**) 0, (**b**) 6.26, (**c**) 12.32, and (**d**) 20.54.

**Figure 6 nanomaterials-14-00814-f006:**
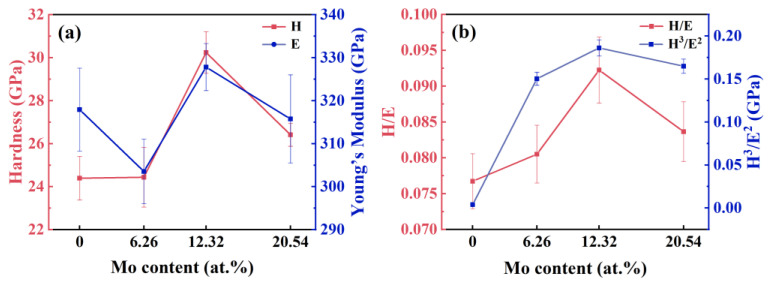
Nano-mechanical properties of (AlCrTiZrMo_x_)N coatings. (**a**) Hardness and Young’s modulus, (**b**) H/E and H^3^/E^2^.

**Figure 7 nanomaterials-14-00814-f007:**
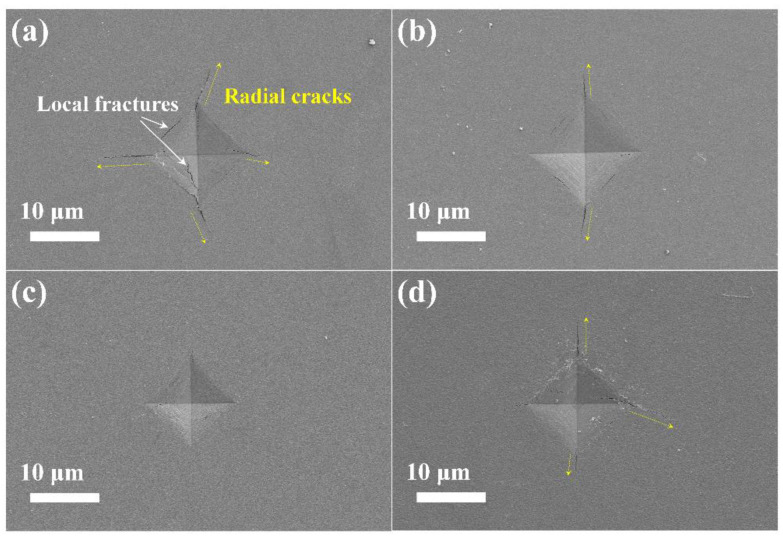
The FESEM of the Vickers indentations for the (AlCrTiZrMo_x_)N coatings with different at.% Mo: (**a**) 0, (**b**) 6.26, (**c**) 12.32, and (**d**) 20.54.

**Figure 8 nanomaterials-14-00814-f008:**
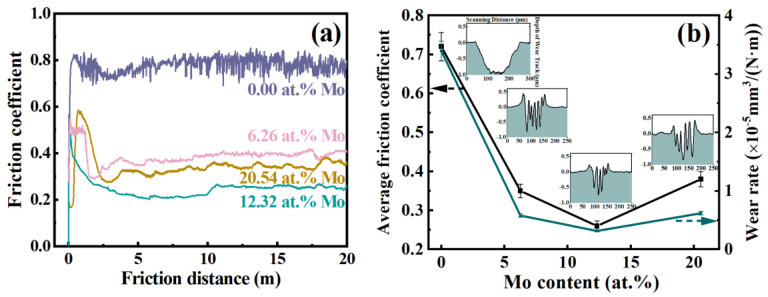
(**a**) Specific COF curve as a function of distance, (**b**) average COF and wear rate. (The illustration is the cross-sectional profiles of the wear track).

**Figure 9 nanomaterials-14-00814-f009:**
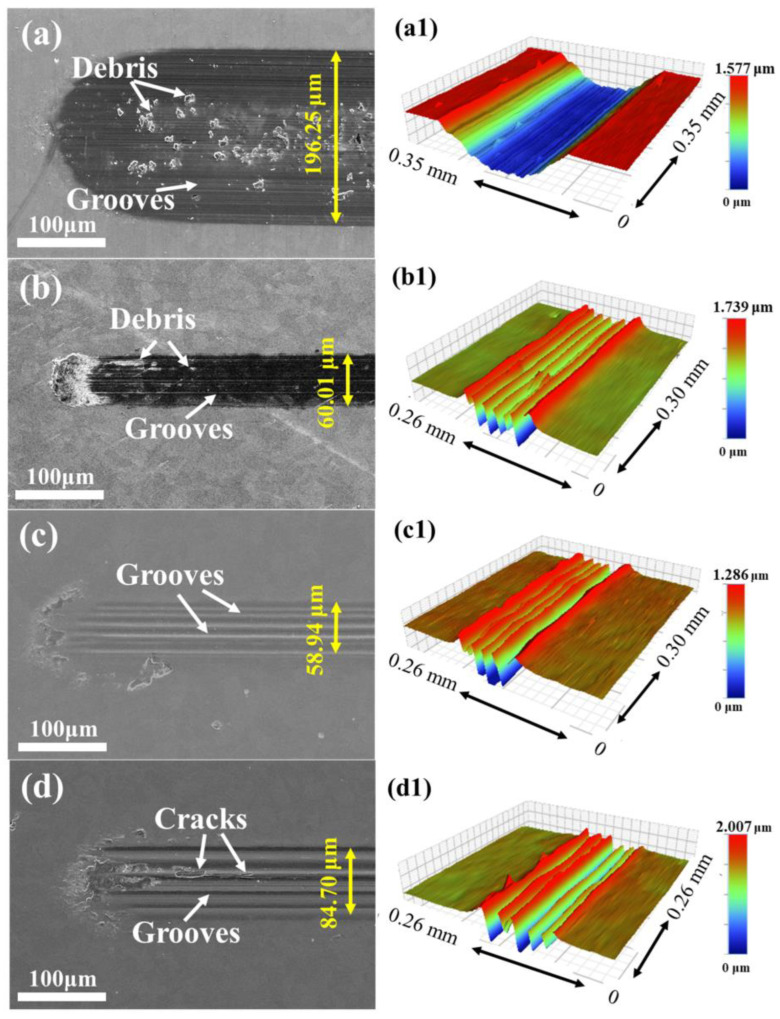
SEM images and 3D morphologies of wear track: (**a**,**a1**) 0, (**b**,**b1**) 6.26, (**c**,**c1**) 12.32, and (**d**,**d1**) 20.54 at.% Mo.

**Figure 10 nanomaterials-14-00814-f010:**
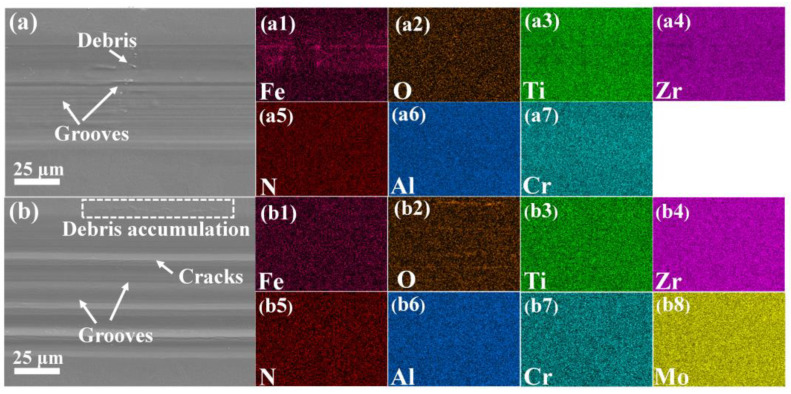
SEM images and surface element mapping of wear track: (**a**,**a1**–**a7**) 0, and (**b**,**b1**–**b8**) 12.32 at.% Mo.

**Figure 11 nanomaterials-14-00814-f011:**
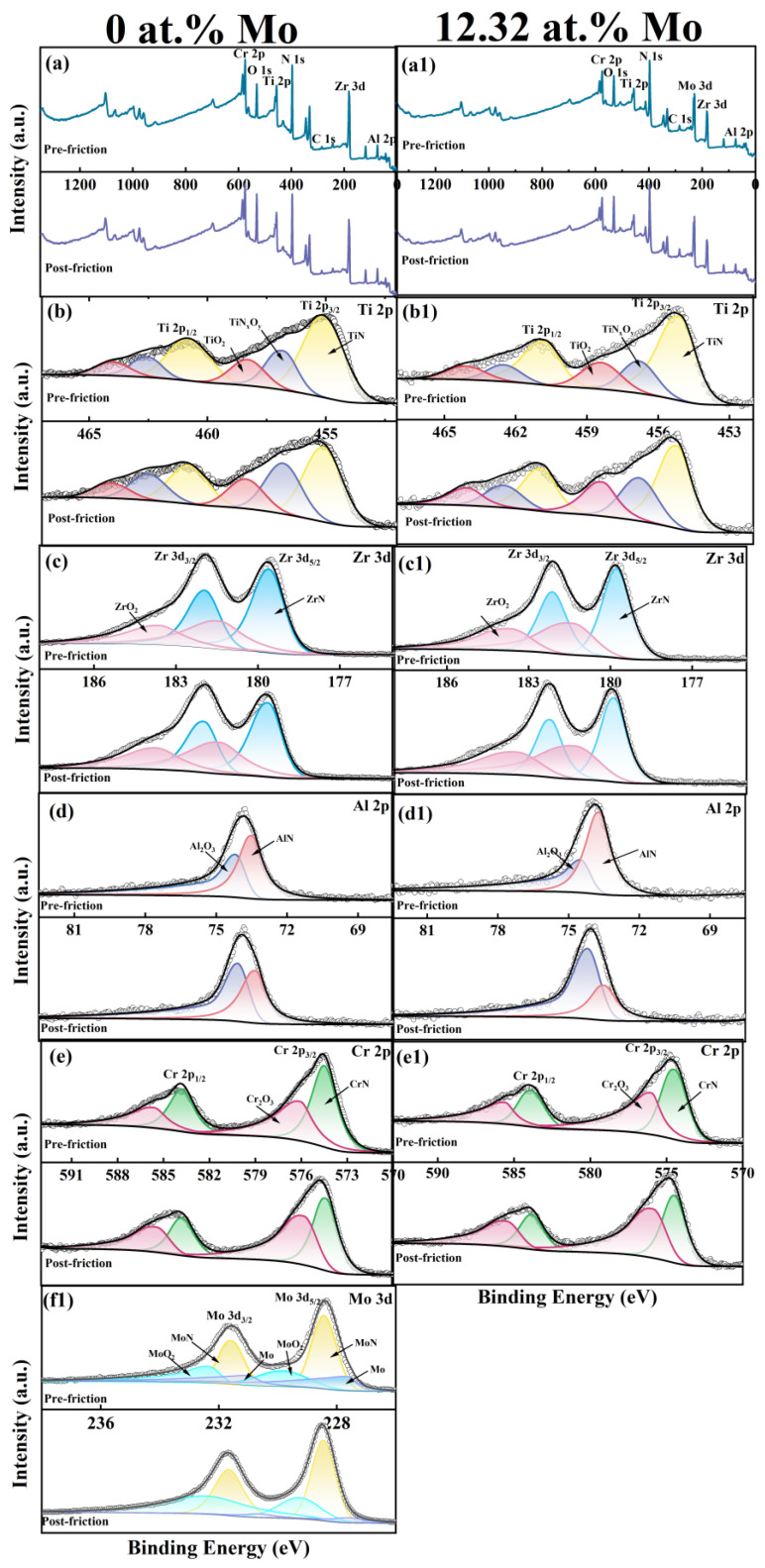
XPS result of (**a**,**a1**) survey spectra, (**b**,**b1**) Ti 2p, (**c**,**c1**) Zr 3d, (**d**,**d1**) Al 2p, (**e**,**e1**) Cr 2p, and (**f1**) Mo 3d fine spectra of the coatings.(The black is the fitting curve, and the other colors are the fitting peaks.)

**Figure 12 nanomaterials-14-00814-f012:**
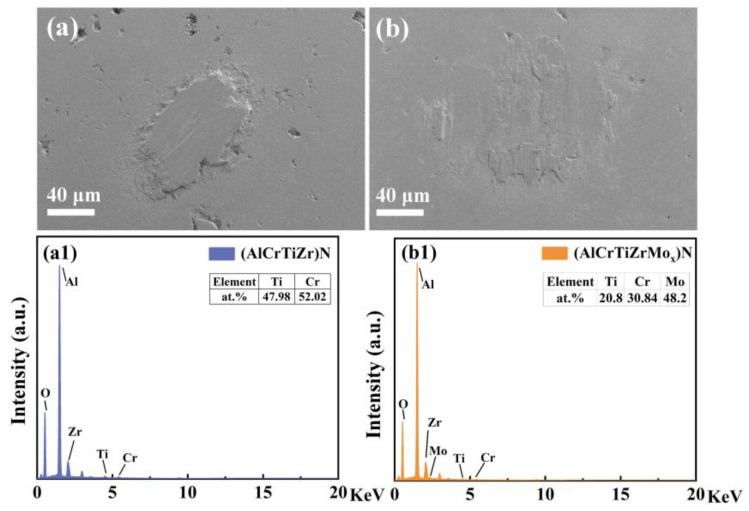
SEM image of transfer films on the ball counterparts against coatings, and the corresponding EDS spectra accompanied with the element content: (**a**,**a1**) 0, and (**b**,**b1**) 12.32 at.% Mo.

**Figure 13 nanomaterials-14-00814-f013:**
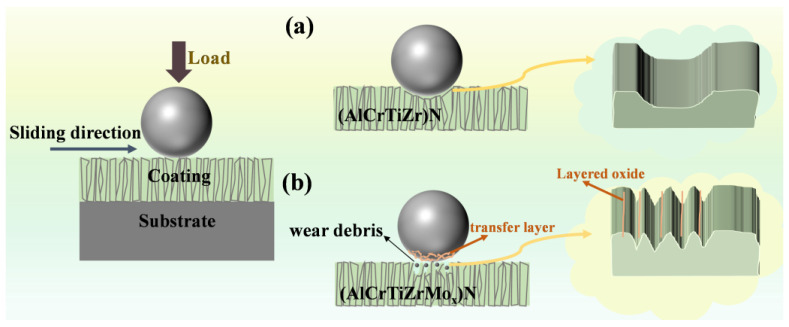
Schematic diagram of the lubrication mechanism: (**a**) (AlCrTiZr)N, and (**b**) (AlCrTiZrMo_x_)N coatings.

**Table 1 nanomaterials-14-00814-t001:** Deposition parameters of (AlCrTiZrMo_x_)N coatings.

DC Power of Target (W)			Deposition Time	RF Bias Power (W)	N_2_ Flow Rate (sccm)	Ar Flow Rate (sccm)
TiZr 200	AlCr 150	Mo 0/30/50/100	240/230 /210/190	30	12	40

**Table 2 nanomaterials-14-00814-t002:** Crystallographic parameters, mechanical properties, and tribological properties of (AlCrTiZrMo_x_)N.

Mo Content	d (nm)	a (Å)	H (GPa)	E (GPa)	H/E	H^3^/E^2^ (GPa)	μ	W_r_ (mm^3^/(N•m))
0 at.%	6.5	4.2849	24.39	317.91	0.077	0.144	0.72	3.39 × 10^−5^
6.26 at.%	7.1	4.3022	24.44	303.56	0.081	0.158	0.35	5.78 × 10^−6^
12.32 at.%	6.3	4.3138	30.24	327.83	0.092	0.257	0.26	3.15 × 10^−6^
20.54 at.%	5.8	4.3010	26.41	315.76	0.084	0.185	0.38	6.18 × 10^−6^

**Table 3 nanomaterials-14-00814-t003:** The atomic proportion of metal oxides of (AlCrTiZrMo_x_)N coatings.

(TiZrAlCrMo_x_)N	Ti (at.%)		Zr (at.%)	Al (at.%)	Cr (at.%)	Mo (at.%)
	Ti-N-O	Ti-O	Zr-O	Al-O	Cr-O	Mo-O
0 at.% Mo-pre	22.64	16.15	53.1	46.34	45.73	
0 at.% Mo-post	31.74	18.56	45.48	55.48	53.23	
12.32 at.% Mo-pre	19.78	18.49	50.01	39.84	50.93	28.10
12.32 at.% Mo-post	28.48	21.35	49.99	67.45	57.39	41.59

## Data Availability

The data presented in this study are available upon request from the corresponding author.

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
