# Peer review of "Study on Microstructure and Tribological Mechanism of Mo Incorporated (AlCrTiZr)N High-Entropy Ceramics Coatings Prepared by Magnetron Sputtering"

_nanomaterials, 2024, doi:10.3390/nano14100814_

Round 1

Reviewer 1 Report

Comments and Suggestions for Authors

The manuscript titled "Study on microstructure and tribological mechanism of Mo incorporated (AlCrTiZr)N high-entropy ceramics coatings prepared by magnetron sputtering" presents a comprehensive investigation into the influence of Mo content on the properties of (AlCrTiZrMox)N coatings. The study explores microstructural changes, hardness, fracture toughness, and tribological behavior, providing valuable insights into the performance of these coatings.

The experimental methodology is meticulously described, highlighting the preparation process of the coatings and the techniques employed for characterization. The results section is well-organized, presenting clear observations regarding the effect of Mo content on various coating properties. The authors effectively demonstrate the relationship between Mo content and changes in preferred orientation, columnar structure, hardness, and friction behavior.

One notable finding is the reduction in friction coefficient and wear rate with increasing Mo content. The formation of a MoO2 lubricating layer during the friction process is particularly intriguing and provides valuable understanding of the tribological mechanism of these coatings. The discussion of the interplay between hardness, fracture toughness, and wear resistance is insightful and enhances the overall contribution of the study to the field of materials science.

However, it should be noted that the manuscript contains an ERROR ("Error! Referecne sourse not found" - lines: 82, 131, 143, 159, 164, 172, 180, 194, 199, 212, 230, 235, 245, 259, 274, 287, 289, 292, 294, 295, 300, 303, 317, 323, 330, 336, 341, 344, 347, 349, 364)  regarding the numbering of cited figures and tables.

It is generally straightforward to infer the referenced figure or table from the descriptive text. but  Figure 13 i am not sure it is referenced.

Additionally, Figures 9 and 10 suffer from too small font sizes, which may hinder the interpretation of the data presented. Addressing these issues would enhance the readability and clarity of the manuscript.

In conclusion, the study presents valuable contributions to the understanding of high-entropy ceramics coatings' tribological behavior. Despite minor flaws, the research is well-conducted and thoroughly analyzed. Therefore, I recommend accepting this manuscript for publication in the journal "Nanomaterials."

Reviewer 2 Report

Comments and Suggestions for Authors

The article is devoted to the deposition and study of wear-resistant ceramic coatings based on solid solutions of metal nitrides. It has been established that the addition of Mo to the ceramic composition significantly improves the properties of the resulting films. The optimal Mo content was determined to provide the best wear resistance. The experimental results are supported by well-founded conclusions. The article can be accepted for publication after minor corrections, described in detail below.

1.      Page 2, section 2.1. What kind of power did magnetron sources have (DC, PF, pulsed)? Was the substrate additionally heated during deposition? Add deposition time in manuscript, please.

2.      Page 2, line 82. Check the reference link, please. There are a lot of similar issues across the manuscript.

3.      Page 7, line 217. The hardness of the samples differs within the limits of error, so it is somewhat incorrect to talk about a trend towards its decrease.

Reviewer 3 Report

Comments and Suggestions for Authors

The presented paper is devoted to a comprehensive study of the structure of Mo incorporated (AlCrTiZr)N high - entropy ceramics coatings at different scale levels, as well as mechanical properties of the composite.

The work was performed at a high professional level using modern analytical methods and equipment. The research is thoroughly performed, the interpretation of the results and their reliability do not cause doubts.

Nevertheless, after reading there are a number of remarks, after correction of which the article certainly deserves to be published:

1. Under (1) the authors cite the Scherrer equation, in which D - is the crystallite size, not the grain size. Later in the text this misconception is repeated. The authors should pay more attention to this issue, since the conclusions drawn regarding the Hall-Petch dependence are correct, but based on a somewhat incomplete understanding of the physical input data.

2 The text of the paper is replete with missing literature references, instead of which the following is presented: “Error! Reference source not found”.

3. A number of figures are difficult to read because of the size. For example, Figure 8, Figure 9, etc.

4. The list of references needs to be supplemented with more relevant sources.

Comments on the Quality of English Language

There are a number of typos in the text. Careful proofreading is necessary.
